# Tumor Heterogeneity of Breast Cancer Assessed with Computed Tomography Texture Analysis: Association with Disease-Free Survival and Clinicopathological Prognostic Factor

**DOI:** 10.3390/diagnostics13233569

**Published:** 2023-11-29

**Authors:** Hyeongyu Yoo, Kyu Ran Cho, Sung Eun Song, Yongwon Cho, Seung Pil Jung, Kihoon Sung

**Affiliations:** 1Department of Radiology, Korea University Anam Hospital, Korea University College of Medicine, Seoul 02841, Republic of Korea; twm0822@naver.com (H.Y.); krcho@korea.ac.kr (K.R.C.); dragon1@korea.ac.kr (Y.C.); 2Department of Surgery, Korea University Anam Hospital, Korea University College of Medicine, Seoul 02841, Republic of Korea; jungspil@gmail.com; 3Technological Lab, Kakaohealthcare, Seongnam-si 13529, Republic of Korea; brown.sung@kakaohealthcare.com

**Keywords:** breast neoplasms, computed tomography, texture analysis, prognosis, disease-free survival

## Abstract

Breast cancer is a heterogeneous disease, and computed tomography texture analysis (CTTA), which reflects the tumor heterogeneity, may predict the prognosis. We investigated the usefulness of CTTA for the prediction of disease-free survival (DFS) and prognostic factors in patients with invasive breast cancer. A total of 256 consecutive women who underwent preoperative chest CT and surgery in our institution were included. The Cox proportional hazards model was used to determine the relationship between textural features and DFS. Logistic regression analysis was used to reveal the relationship between textural features and prognostic factors. Of 256 patients, 21 (8.2%) had disease recurrence over a median follow-up of 60 months. For the prediction of shorter DFS, higher histological grade (hazard ratio [HR], 6.12; *p* < 0.001) and lymphovascular invasion (HR, 2.93; *p* = 0.029) showed significance, as well as textural features such as lower mean attenuation (HR, 4.71; *p* = 0.003) and higher entropy (HR, 2.77; *p* = 0.036). Lower mean attenuation showed a correlation with higher tumor size, and higher entropy showed correlations with higher tumor size and Ki-67. In conclusion, CTTA-derived textural features can be used as a noninvasive imaging biomarker to predict shorter DFS and prognostic factors in patients with invasive breast cancer.

## 1. Introduction

Breast cancer is a heterogeneous disease, and tumor heterogeneity in breast cancer is divided into intertumoral heterogeneity and intratumoral heterogeneity [1,2,3,4,5,6,7]. Intratumoral heterogeneity, which can be different according to genetic and phenotypic variations, has been a major obstacle to effective personalized treatment in breast cancer patients, as it drives metastasis, progression, and treatment resistance [8]. According to previous studies, patients with high intratumoral heterogeneity have a poorer prognosis than those with low intratumoral heterogeneity [9,10,11]. For the evaluation of intratumoral heterogeneity, assessment of both genetic and phenotypic variation is essential. Genetic assessment is performed using partial tissue samples derived from surgical or biopsy samples. Due to heterogeneity in the structure of breast cancer, genetic assessment may not be sufficient for assessing intratumoral heterogeneity [12]. Therefore, it is clinically important to evaluate the complete phenotypic variation within a tumor using noninvasive methods such as computed tomography (CT) or magnetic resonance imaging (MRI), which can assess the whole tumor phenotype [7].

To evaluate the spatial heterogeneity of tumors, computed tomography texture analysis (CTTA), which can quantify the intensity of pixel histograms of tumors and reflect intratumoral heterogeneity, has been widely used [13,14,15,16,17,18]. In a variety of tumor types, CTTA can predict pathological features, response to therapy, and prognosis [14,15,16,17,18]. However, there have been no published studies on the use of CTTA in breast cancer patients because chest CT is not the primary method for the evaluation of breast cancer, and current guidelines do not recommend chest CT in patients with early breast cancer [19,20]. Nevertheless, chest CT has been widely and steadily used in clinical practice because it is useful for the accurate evaluation of asymptomatic metastasis to the lungs, bone, lower part of the liver, and axillary and supraclavicular lymph nodes (LN) [21]. Also, chest CT can be used as a staging work-up method for patients who cannot undergo breast MRI due to obesity, MRI contrast allergy, renal insufficiency, or claustrophobia. In the United States, 27% (36.2% of patients with stage II tumors and 11% of patients with stage I tumors) of patients with early breast cancer underwent chest CT scans, and in Australia, 40% of patients underwent chest CT [19,22]. In South Korea, 97.2% of patients underwent chest CT [23]. Therefore, chest CT-derived textural features can be used for predicting survival outcomes or prognoses in patients with invasive breast cancer.

The purpose of this study was to investigate the usefulness of chest CT-derived textural features in predicting disease-free survival (DFS) and to investigate the association between textural features and clinicopathological prognostic factors in patients with invasive breast cancer. 

## 2. Materials and Methods

### 2.1. Patients

The institutional review board approval was acquired for this retrospective study, and the need for written informed patient consent was waived. Through a review of medical records between January 2013 and May 2015, we identified consecutive women who had been newly diagnosed with primary invasive breast cancers. The inclusion criteria were as follows: (a) pretreatment postcontrast chest CT performed at our institution, (b) breast surgery such as mastectomy or breast-conserving therapy performed at our institution, and (c) lesion visible on chest CT correlated with those identified on preoperative breast MRI. The exclusion criteria were as follows: (a) neoadjuvant chemotherapy (NAC) before surgery, (b) distant metastasis at the time of diagnosis, (c) absent histological grade, and (d) patient who had undergone mammoplasty. 

### 2.2. CT Acquisition 

For the evaluation of the involvement of LNs or distant organs such as lungs or bones, chest CTs were performed for all patients with a supine position using various CT scanners, including Brilliance 64, Ingenuity 128 (Philips Medical Systems, Amsterdam, The Netherlands), and Somatom Definition Flash (Siemens, Erlangen, Germany). Our chest CT protocols included a postcontrast scan (280 ms gantry rotation, 80–120 kVp with 30–300 mAs, 1 mm slice collimation) according to the automatic tube current modulation. The reconstruction parameters comprised a 3 mm slice thickness and no reconstruction interval. Intravenous contrast medium (100 mL of iopromide (ProsureM300, LG Life Sciences, Seoul, Republic of Korea) was injected at a rate of 2.5–3 mL/s, followed by a 40 mL saline bolus. Scanning was commenced 90 s after contrast medium injection.

### 2.3. Texture Analysis 

Preoperative CT images were retrospectively reviewed by a radiologist with 11 years of experience in breast imaging (S.E.S) who was blinded to the clinicopathological findings. With a reference standard of breast MRI images, texture analysis was performed by drawing a region of interest (ROI) around the entire enhancing tumor margin at the maximum cross-sectional area of the tumor on axial postcontrast DICOM images with a mediastinal window setting using commercial software (TexRAD software VER 3.9, Feedback Medical Ltd., Cambridge, UK), which is a postprocessing software that uses a filtration-histogram technique (Figure 1 and Figure 2). In the case of multifocal or multicentric diseases, the largest visible tumor was selected to draw the ROI as an index. Texture quantification of the ROI was performed with each filter setting, i.e., a Laplacian of Gaussian spatial bandpass filter, including fine (spatial scaling factor [SSF] 2), medium (SSF 3–4), and coarse (SSF 5–6) filter settings. A fine filter value emphasizes the fine anatomic details, whereas a coarse filter value emphasizes the coarse features [24]. Various texture parameters were calculated. Mean attenuation indicates the average attenuation value. Standard deviation (SD) shows the degree of dispersion from the average. The mean of positive pixels (MPP) indicates the average gray-level intensity above the threshold of zero. Entropy shows irregularity or complexity of pixel intensities. Kurtosis indicates the pointiness of the pixel distribution, and skewness represents its degree of asymmetry.

### 2.4. Clinicopathological Data 

The final histopathological results were obtained from surgical specimens to determine the histological type, histological grade, tumor size, presence of LN involvement and lymphovascular invasion, and estrogen receptor (ER), progesterone receptor (PR), human epidermal growth factor receptor 2 (HER2), and Ki-67 status. ER and PR positivity was determined using a cutoff value of >1% positively stained nuclei. The HER2 staining intensity was scored as 0, 1+, 2+, and 3+, and tumors with HER2 scores of 3+ were considered positive. Tumors with HER2 scores of 2+ were further evaluated with silver-enhanced in situ hybridization. Based on the results of ER, PR, and HER2 analyses, the tumors were characterized into three molecular subtypes—luminal-like (ER/PR-positive and HER2-negative), HER2-like (ER/PR-negative and HER2-positive), and basal-like (ER/PR/HER2-negative) tumors [25]. Regarding the Ki-67 expression status, nuclear staining ≥14% was considered a high level of expression.

### 2.5. Statistical Analysis

The primary endpoint was DFS, which was calculated as the time interval from the date of surgery to the date of the first event, such as breast cancer recurrence (locoregional or distant recurrence) or the development of new primary contralateral breast cancer. Patients were followed up until the first event or until May 2020 if they were alive. 

The clinicopathological characteristics and CT texture features were compared between patients with and without recurrence using the chi-square test, Mann–Whitney U test, or Student’s *t*-test. For survival analysis, significant CT texture features were dichotomized according to the optimal cutoff values identified by receiver operating characteristic (ROC) curve analysis using the maximum Youden index. According to the cutoff values, patients were categorized into a low-risk recurrence group or a high-risk recurrence group. Kaplan–Meier survival curves were drawn to compare survival between the high-risk and low-risk recurrence groups. To determine the relationship between clinicopathological or textural features and DFS, the Cox proportional hazards model was used. Features with *p*-values < 0.05 at univariate analysis were further analyzed using a multivariate analysis. To adjust for multiple comparisons, we performed a false discovery rate correction using the Benjamini and Hochberg method. Adjusted *p* < 0.05 was considered indicative of a significant difference. The relationships between textural features associated with DFS and clinicopathological prognostic factors were assessed using the Mann–Whitney U test or Student’s *t*-test and linear logistic regression analysis (SPSS software, version 20.0; IBM Corp., Armonk, NY, USA). *p*-values < 0.05 were considered statistically significant.

## 3. Results

### 3.1. Patient Characteristics and Disease-Free Survival

Among 350 consecutive women who had been newly diagnosed with primary invasive breast cancers, 41 patients who had undergone NAC before surgery because clinicopathologic prognostic factor could be changed after NAC, 39 patients who had previously undergone excision, 9 patients who had distant metastasis at the time of diagnosis, 4 patients who had no histological grade available, and one patient who had undergone mammoplasty before surgery, were excluded. Finally, 256 patients were enrolled (Figure 3). The mean interval between preoperative CT and surgery was 9.5 days. Of the 256 patients (mean age, 54 ± 11 years; range, 30–87 years), 176 (68.8%) patients underwent breast-conserving surgery and 80 (31.2%) patients underwent mastectomy. One patient who underwent breast-conserving surgery had a positive resection margin on the pathologic report. 

Of the 256 patients (mean age, 54 ± 11 years; range, 30–87 years), the most common histological types were invasive ductal carcinoma (*n* = 218), invasive lobular carcinoma (*n* = 13), mucinous carcinoma (*n* = 7), other or mixed type (*n* = 5), medullary carcinoma (*n* = 4), tubular carcinoma (*n* = 3), papillary carcinoma (*n* = 3), and metaplastic carcinoma (*n* = 3). The molecular subtypes were luminal-like (*n* = 167), HER2-like (*n* = 50), and basal-like subtypes (*n* = 39).

There were 21 (nine locoregional, nine distant, and three contralateral) events and three deaths. The median follow-up period was 60 months (range, 11–89 months). The mean time to events was 39 months (range, 11–72 months). Patients with recurrence were more likely to have a higher histological grade, tumor size >2 cm, lymphovascular invasion, and a high Ki-67 expression than patients without recurrence (Table 1).

### 3.2. CT Texture Features

The CT textural features were compared between the nonrecurrence group (*n* = 235) and the recurrence group (*n* = 21). When no image filtration (SSF0) was used, textural features were not significantly different between the two groups. When images were obtained using SSF2–6, the recurrence group had a lower mean CT attenuation and MPP than the nonrecurrence group. When using SSF2, the recurrence group had higher entropy than the nonrecurrence group, and when using SSF6, the recurrence group had higher kurtosis than the nonrecurrence group (Table 2). The optimal cutoff values for CT textural features were acquired. At SSF2, the optimal cutoff value was 32.76 Hounsfield units (HU) for mean, 68.06 for MPP, and 5.31 for entropy. At SSF3, the optimal cutoff value was 47.31 HU for mean and 66.74 for MPP. At SSF4, the optimal cutoff value was 54.97 HU for mean and 80.81 for MPP. At SSF5, the optimal cutoff value was 62.59 HU for mean and 74.50 for MPP. At SSF6, the optimal cutoff value was 52.45 HU for mean, 86.40 HU for MPP, and −0.90 for kurtosis. 

### 3.3. Survival Analysis Using Cox Proportional Hazard Models

In the univariate Cox proportional hazards model, the clinicopathological features of higher histological grade, larger tumor size, presence of lymphovascular invasion, and higher Ki-67 status were associated with shorter DFS outcomes. For CT textural features, lower MPP and higher entropy using SSF2, lower mean and lower MPP using SSF3, lower mean and lower MPP using SSF4, lower mean and lower MPP using SSF5, and lower mean and lower MPP using SSF6 were associated with shorter DFS outcomes (Table 3). In multivariate analysis, higher histological grade (HR, 6.12; *p* < 0.001), presence of lymphovascular invasion (HR, 2.93, *p* = 0.029), the textural feature of lower mean attenuation using SSF5 (HR, 4.71; *p* = 0.003), and higher entropy using SSF2 (HR, 2.77; *p* = 0.036) were significant factors for predicting shorter DFS (Table 4). 

### 3.4. Association with Textural Features and Clinicopathological Features 

At univariate analysis, lower mean attenuation at SSF5 was associated with higher tumor size, whereas higher entropy at SSF2 was correlated with higher histologic grade, higher tumor size, presence of LN metastasis and lymphovascular invasion, negative HER2 status, and higher Ki-67 status (Table 5). At multivariate analysis, higher entropy at SSF2 was correlated with higher tumor size (odds ratio [OR] = 6.59; *p* < 0.001) and higher Ki-67 status (OR = 1.94; *p* = 0.049) (Table 6). Kaplan–Meier survival curves using lower mean attenuation with SSF5 or higher entropy with SSF2 were drawn (Figure 4).

## 4. Discussion

Intratumoral heterogeneity evolves unpredictably during disease progression, posing significant challenges for chemotherapeutics and necessitating a paradigm shift from standard pathological classifications of breast cancers to a more personalized approach in which intratumoral heterogeneity is thoroughly characterized prior to treatment [3,5]. Recent radiomic studies have focused on the impact of intratumoral heterogeneity on the prognosis of patients with breast cancer. Son et al. [26] revealed that intratumoral metabolic heterogeneity assessed using 18F-fluorodeoxyglucose positron emission tomography (PET)/CT was a prognostic factor for overall survival. Two recent articles using breast MRI also demonstrated that higher values of kinetic heterogeneity using computer-aided diagnosis or a higher apparent diffusion coefficient difference value at diffusion-weighted MRI showed associations with poor distant metastasis-free survival in patients with invasive breast cancers [11,27]. 

Texture analysis to quantify the spatial pattern of pixel intensities of whole tumors is an objective measure of intratumoral heterogeneity [14,15,16,17]. In patients with breast cancer, several studies have used texture analysis based on breast MRI for detecting microcalcifications [28], distinguishing between breast cancer subtypes [29,30], differentiating between benign and malignant lesions [31,32], and predicting outcomes in patients with neoadjuvant chemotherapy [33,34]. For the prediction of survival outcomes, two studies using breast MRI revealed that the textural feature of entropy was associated with survival outcomes in patients with breast cancer [35,36]. However, there are no published studies on the use of CTTA in patients with breast cancer.

Among texture features, the most important feature that showed an association with DFS in our study was the mean attenuation value of the tumor. In the recurrence group, the mean attenuation of the tumor using all SSFs was significantly lower than that in the nonrecurrence group. The reason why lower mean CT attenuation was associated with poor DFS in our study could be explained by the results of previous studies [37,38], which investigated the correlation between the degree of necrosis on CT and the expression of hypoxic and angiogenesis biomarkers in breast cancer. In that study, lower CT attenuation of breast cancer was significantly associated with tumor necrosis and hypoxic biomarkers, which appears to be strongly associated with malignant progression, tumor propagation, and treatment resistance [39]. Several CTTA studies also reported that lower mean CT attenuation of tumors predicted poor survival outcomes in metastatic renal cell carcinoma [40] and pancreatic cancer [41,42]. Based on our study results, a lower mean CT attenuation of the tumor of <62.59 HU at SSF5 on chest CT could be useful as an imaging biomarker for predicting poor DFS in patients with primary breast cancer.

Entropy has been recognized as a marker of tumor heterogeneity. Our study showed that entropy >5.31 using SSF2 is an independent risk factor for poor DFS. Previous CTTA studies of head and neck squamous cancer, esophageal cancer, and non-small-cell lung cancer also revealed that higher entropy was associated with a poor prognosis [14,17,18]. Two previous studies using breast MRI also reported that entropy was the most important texture feature for predicting the prognosis of breast cancer [29,35]. Waugh et al. [29] demonstrated that entropy on breast MRI could differentiate histological and immunohistochemical subtypes in breast cancer. Kim et al. [35] also showed that higher entropy on T2-weighted breast MRI and lower entropy on postcontrast T1-weighted breast MRI were associated with poor recurrence-free survival in patients with invasive breast cancer. Collectively, our results and previous study results suggest that textural features of higher entropy can be used for predicting poor prognosis in patients with invasive breast cancers, regardless of the use of CT or MRI.

Among the clinicopathological features, the presence of lymphovascular invasion and higher histological grade were significantly associated with recurrence in our study. In a previous study, the presence of lymphovascular invasion was an important prognostic factor independent of LN status, histological grade, ER status, and tumor size [43]. In another study, higher histological grade was a prognostic factor, independent of the number of positive LNs and tumor size [44]. Truong et al. [45] also found that the presence of lymphovascular invasion, in combination with histological grade III, increased the risk of locoregional recurrence, which was consistent with our study results. 

This study has several limitations. First, this was a retrospective study, and selection bias might have been present in patient enrollment. Second, we used four different types of CT scanners using nonuniform CT acquisition factors. However, Miles et al. showed that texture parameters are less sensitive to differing tube voltages and currents [44]. Third, our study was performed at a single institution, and no external validation was performed. Fourth, a single user manually performed segmentation for CTTA. Thus, interobserver variability was not evaluated. However, a previous study has revealed that the intra-observer reproducibility of single-section measurements is quite high [45]. Fifth, contrast enhancement of breast cancer on a postcontrast chest CT scan can depend on many individual factors, such as age, sex, body weight and height, cardiac output, renal function, and hydration status, resulting in variability in CTTA [46]. 

## 5. Conclusions

In conclusion, textural features such as lower mean attenuation and higher entropy using preoperative chest CT were associated with shorter DFS in patients with invasive breast cancer. In regards to prognostic factors, lower mean attenuation may reflect higher tumor size, and higher entropy may reflect higher tumor size and higher Ki-67. From our exploratory study results, CTTA-derived tumor heterogeneity can be used as a noninvasive imaging biomarker to provide additional methods for risk stratification in patients with invasive breast cancers who cannot undergo breast MRI due to obesity, MRI contrast allergy, renal insufficiency, or claustrophobia.

## Figures and Tables

**Figure 1 diagnostics-13-03569-f001:**
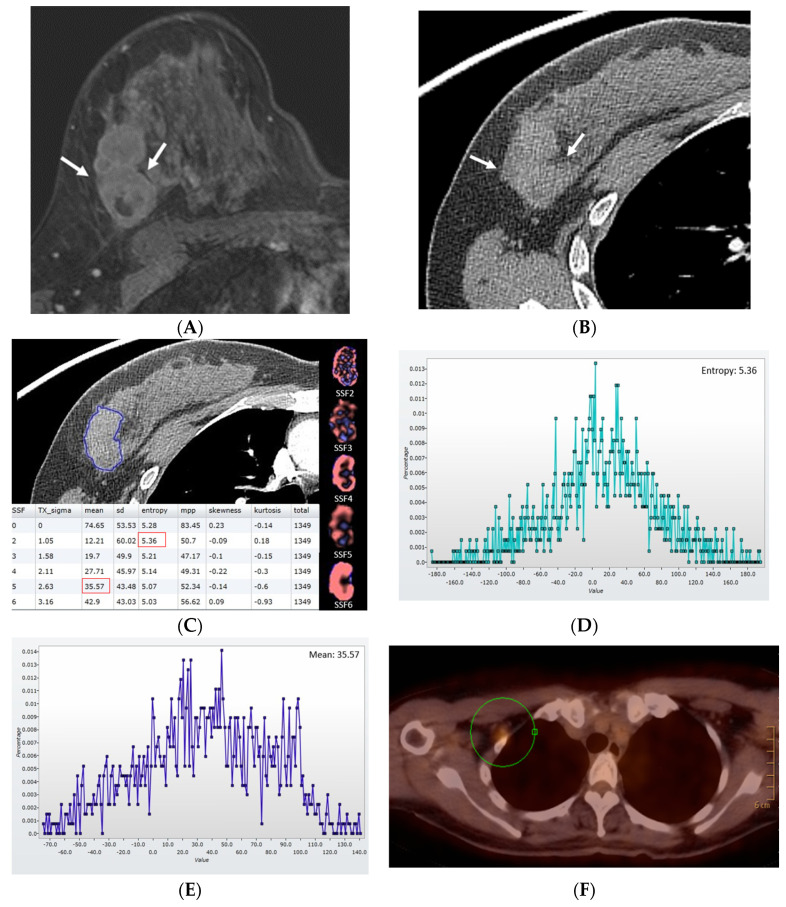
Findings in a 45-year-old woman with basal-like cancer in the right breast (pathological stage IIA, T2N0, histological grade III). (**A**) Axial contrast-enhanced T1−weighted MRI image acquired for 2 min shows an enhancing mass (arrows) in the right breast. (**B**) Axial postcontrast CT image shows an enhancing mass with a central area of low attenuation (arrows) in the right breast. (**C**) With the reference standard of an MRI image, a region of interest was drawn along the margin of the enhancing tumor on the CT image (blue line), and texture parameters without image filtration (SSF0) and with image filtration (SSF2−6) were acquired. Mean attenuation using SSF5 was 35.57, and entropy using SSF2 was 5.36. (**D**) CT texture histogram using SSF5 provides information on mean attenuation. (**E**) CT texture histogram using SSF2 provides information on entropy. (**F**) Ipsilateral lymph node metastasis (circle) in her axilla on PEC−CT 1 year after surgery.

**Figure 2 diagnostics-13-03569-f002:**
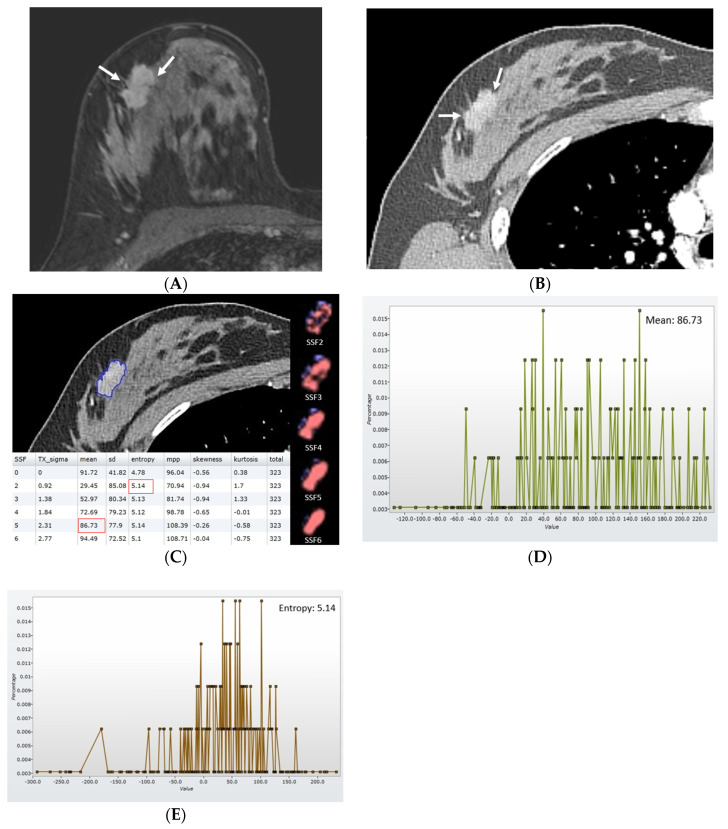
Findings in a 42−year-old woman with luminal-like cancer in the right breast (pathological stage IIA, T2N0, histological grade I). (**A**) Axial contrast-enhanced T1−weighted MRI image acquired for 2 min shows an enhancing mass in the right breast. (**B**) Axial postcontrast CT image shows an enhancing mass without a central area of low attenuation in the right breast. (**C**) With the reference standard of an MRI image, a region of interest was drawn along the margin of enhancing tumor on CT image (blue line), and texture parameters without image filtration (SSF0) and with image filtration (SSF2−6) were acquired. Mean attenuation using SSF5 was 86.73, and entropy using SSF2 was 5.14. (**D**) CT texture histogram using SSF5 provides information on mean attenuation. (**E**) CT texture histogram using SSF2 provides information on entropy. There have been no signs of recurrence for 5 years since the surgery.

**Figure 3 diagnostics-13-03569-f003:**
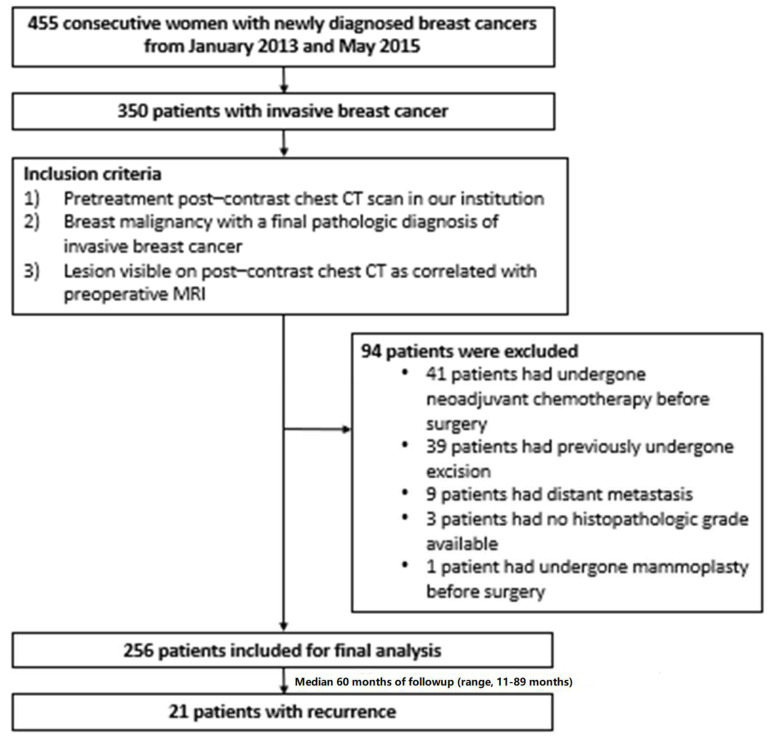
Flowchart of patient inclusion.

**Figure 4 diagnostics-13-03569-f004:**
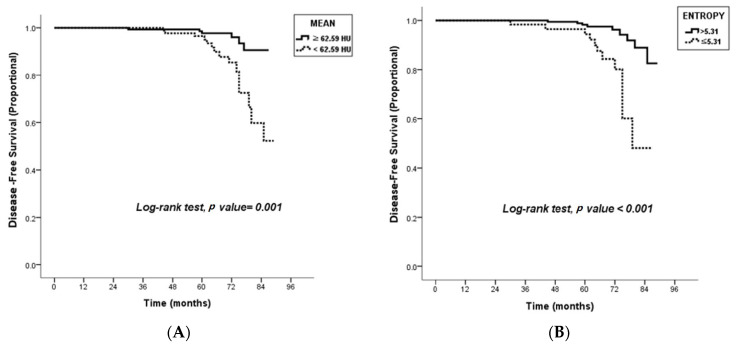
Kaplan–Meier curves. There are significant differences in disease-free survival according to a mean attenuation of tumor of 62.59 Hounsfield units using SSF5 (*p* = 0.001) (**A**) and an entropy of 5.31 (*p* < 0.006) (**B**) between the recurrence group and nonrecurrence group.

**Table 1 diagnostics-13-03569-t001:** Patient characteristics according to recurrence status.

Features	All Patients (*n* = 256) *	Nonrecurrence Group (*n* = 235)	Recurrence Group (*n* = 21)	*p*-Value
Patient age (years)	54.86 ± 11.34	54.82 ± 11.10	55.28 ± 14.06	0.860
Histological type				1.000
Invasive ductal	218 (85.1)	200 (85.1)	18 (85.7)	
Others	38 (14.9)	35 (14.8)	3 (14.3)	
Histological grade				<0.001
I or II	180 (70.3)	173 (73.6)	7 (33.3)	
III	76 (29.7)	62 (26.4)	14 (66.7)	
Tumor size				0.006
≤2 cm	157 (61.3)	150 (63.8)	7 (33.3)	
>2 cm	99 (38.7)	85 (36.2)	14 (66.7)	
LN ^a^ status				0.063
Negative	169 (66.0)	159 (67.7)	10 (47.6)	
Positive	87 (34.0)	76 (32.3)	11 (52.4)	
Lymphovascular invasion				0.013
Absent	218 (85.2)	204 (86.8)	14 (66.7)	
Present	38 (14.8)	31 (13.2)	7 (33.3)	
ER ^b^ status				0.069
Negative	57 (22.3)	49 (20.9)	8 (38.1)	
Positive	199 (77.7)	186 (79.1)	13 (61.9)	
PR ^c^ status				0.230
Negative	69 (27.0)	61 (26.0)	8 (38.1)	
Positive	187 (73.0)	174 (74.0)	13 (61.9)	
HER2 ^d^ status				1.000
Negative	206 (80.5)	189 (80.4)	17 (81.0)	
Positive	50 (19.5)	46 (19.6)	4 (19.0)	
Molecular subtype				0.197
Luminal-like	167 (65.2)	156 (66.4)	11 (52.4)	
HER2-like	50 (19.5)	46 (19.6)	4 (19.0)	
Basal-like	39 (15.2)	33 (14.0)	6 (28.6)	
Ki-67 status				0.001
Low (<14%)	159 (62.1)	153 (65.1)	6 (28.6)	
High (≥14%)	97 (37.9)	82 (34.9)	15 (71.4)	

* Data are presented as number of cancers with percentages in parentheses. ^a^ LN, lymph node; ^b^ ER, estrogen receptor; ^c^ PR, progesterone receptor; ^d^ HER2, human epidermal growth factor receptor type 2.

**Table 2 diagnostics-13-03569-t002:** CT Textural features according to recurrence status.

CT Textural Features	All Patients (*n* = 256) *	Nonrecurrence Group (*n* = 235)	Recurrence Group (*n* = 21)	*p*-Value	Adjusted *p*-Value ^†^
SSF ^a^ 0					
Mean	84.68 (66.98, 101.34)	85.00 (67.17, 103.04)	82.54 (53.17, 94.06)	0.175	
SD ^b^	47.16 (41.91, 52.78)	47.09 (42.24, 53.10)	47.98 (38.47, 50.17)	0.339	
MPP ^c^	90.02 (73.36, 105.63)	89.91 (73.75, 106.77)	90.20 (63.92, 96.75)	0.157	
Entropy	4.78 (4.50, 5.01)	4.78 (4.48, 5.01)	4.96 (4.59, 5.16)	0.098	
Skewness	−0.03 (−0.19, 0.13)	−0.04 (−0.21, 0.13)	0.10 (−0.13, 0.17)	0.102	
Kurtosis	−0.09 (−0.31, 0.09)	−0.09 (−0.09, 0.10)	−0.05 (−0.27, 0.02)	0.802	
SSF2					
Mean	40.44 (29.00, 65.40)	42.78 (29.45, 67.47)	32.74 (18.89, 43.07)	0.004	0.032
SD	73.52 (63.57, 88.54)	74.10 (63.67, 88.62)	65.68 (61.30, 88.52)	0.395	
MPP	78.55 (63.68, 100.52)	79.68 (64.16, 101.72)	64.31 (57.68, 83.45)	0.016	0.039
Entropy	5.06 (4.71, 5.31)	5.05 (1.70, 5.28)	5.32 (4.82, 5.42)	0.039	0.022
Skewness	−0.09 (−0.34, 0.10)	−0.11 (−0.36, 0.10)	−0.04 (−0.60, 0.31)	0.821	
Kurtosis	−0.14 (−0.51, 0.27)	−0.16 (−0.52, 0.25)	0.18 (−0.39, 0.38)	0.322	
SSF3					
Mean	57.10 (−0.51, 0.27)	60.29 (41.44, 94.53)	40.32 (26.46, 58.78)	0.002	0.022
SD	69.06 (55.99, 80.75)	69.08 (56.92, 80.82)	63.91 (51.06, 80.98)	0.265	
MPP	83.67 (66.04, 113.85)	85.01 (68.63, 116.69)	65.78 (53.74, 87.30)	0.002	0.022
Entropy	5.00 (4.67, 5.22)	4.99 (4.67, 5.21)	5.21 (4.65, 5.34)	0.107	
Skewness	−0.18 (−0.44, 0.04)	−0.18 (−0.44, 0.04)	−0.19 (−0.38, 0.03)	0.712	
Kurtosis	−0.36 (−0.72, 0.11)	−0.37 (−0.73, 0.10)	−0.33 (−0.62, 0.36)	0.265	
SSF4					
Mean	70.51 (47.99, 100.00)	72.14 (49.72, 109.77)	48.06 (30.58, 68.78)	0.004	0.032
SD	64.09 (50.95, 80.32)	64.88 (51.67, 80.62)	58.75 (45.97, 79.03)	0.502	
MPP	89.97 (67.77, 122.21)	91.39 (69.00, 123.78)	67.27 (52.29, 90.34)	0.006	0.036
Entropy	4.98 (4.63, 5.19)	4.96 (4.62, 5.19)	5.16 (4.63, 5.30)	0.126	
Skewness	−0.24 (−0.52, −0.01)	−0.24 (−0.52, 0.00)	−0.24 (−0.74, 0.40)	0.862	
Kurtosis	−0.49 (−0.85, −0.12)	−0.51 (−0.87, −0.02)	−0.30 (−0.74, 0.40)	0.061	
SSF5					
Mean	72.55 (47.57, 105.26)	74.22 (50.81, 110.82)	49.97 (31.48, 79.34)	0.004	0.032
SD	60.84 (45.76, 79.58)	61.08 (45.97, 80.23)	57.88 (44.10, 75.15)	0.502	
MPP	88.76 (67.56, 124.56)	91.13 (69.23, 126.48)	68.64 (53.83, 99.96)	0.006	0.036
Entropy	4.94 (4.57, 5.21)	4.92 (4.58, 5.18)	5.11 (4.56, 5.28)	0.126	
Skewness	−0.26 (−0.51, −0.05)	−0.26 (−0.50, −0.05)	−0.29 (−0.62, 0.02)	0.862	
Kurtosis	−0.60 (−0.60, −0.19)	−0.64 (−0.89, −0.22)	−0.40 (−0.71, 0.23)	0.061	
SSF6					
Mean	69.69 (43.71, 102.11)	74.22 (45.80, 104.65)	42.90 (24.56, 77.29)	0.011	0.039
SD	57.27 (41.61, 79.39)	56.63 (41.67, 81.58)	60.03 (40.98, 73.46)	0.777	
MPP	89.16 (59.21, 122.40)	90.03 (61.50, 124.24)	68.90(51.85, 100.47)	0.020	0.039
Entropy	4.92 (4.52, 5.20)	4.90 (4.51, 5.19)	5.03 (4.59, 5.36)	0.113	
Skewness	−0.03 (−0.19, 0.13)	−0.29 (−0.53, −0.07)	−0.41 (−0.67, 0.09)	0.898	
Kurtosis	−0.09 (−0.09, 0.09)	−0.70 (−0.93, −0.31)	−0.27 (−0.76, 0.24)	0.018	0.039

* Data are median with Q1 and Q3 percentiles in parentheses. ^†^ For adjusted *p* values, we performed a false discovery rate correction using the Benjamini and Hochberg method. ^a^ SSF, spatial scale filter; ^b^ SD, standard deviation; ^c^ MPP, mean of positive pixels.

**Table 3 diagnostics-13-03569-t003:** Univariate Cox proportional hazards analysis of variables associated with disease-free survival.

	Univariate Analysis		
Variable	HR	95% CI	*p*-Value
Clinicopathologic features			
Histological grade			0.002
III	4.260	1.713–10.590	
I or II	Reference		
Tumor size			0.008
>2 cm	3.461	1.392–8.606	
≤2 cm	Reference		
Stage			
II	2.398	0.872–6.593	0.090
I	Reference		
Lymphovascular invasion			0.034
Present	2.678	1.077–6.664	
Absent	Reference		
Ki-67 status			0.015
High (≥14%)	3.269	1.263–8.464	
Low (<14%)	Reference		
Textural Features			
Mean using SSF2 ^a^ <32.76 HU ≥32.76 HU	23.267 Reference	0.025–21,555.829	0.367
MPP ^b^ at SSF2			
<68.06 HU	2.395	1.006–5.702	0.048
≥68.06 HU	Reference		
Entropy using SSF2			<0.001
>5.31	4.809	2.007–11.523	
≤5.31	Reference		
Mean using SSF3			0.017
<47.31 HU	2.926	1.209, 7.077	
≥47.31 HU	Reference		
MPP using SSF3			0.001
<66.74 HU	4.597	1.901–11.114	
≥66.74 HU	Reference		
Mean using SSF4			0.001
<54.97 HU	4.401	1.771–10.934	
≥54.97 HU	Reference		
MPP using SSF4			0.004
<80.81 HU	4.023	1.559–10.383	
≥80.81 HU	Reference		
Mean using SSF5			0.002
<62.59 HU	4.390	1.698–11.348	
≥62.59 HU	Reference		
MPP using SSF5			0.005
<74.50 HU	3.530	1.459–8.540	
≥74.50 HU	Reference		
Mean using SSF6			0.012
<52.45 HU	3.057	1.283–7.281	
≥52.45 HU	Reference		
MPP using SSF6			0.021
<86.40 HU	3.070	1.188–7.931	
≥86.40 HU	Reference		
Kurtosis using SSF6			0.168
>−0.50	1.844	0.773–4.399	
≤−0.50	Reference		

^a^ SSF, spatial scale filter; ^b^ MPP, mean of positive pixels.

**Table 4 diagnostics-13-03569-t004:** Multivariate Cox proportional hazards analysis of variables associated with disease-free survival.

	Multivariate Analysis		
Variable	HR	95% CI	*p*-Value
Clinicopathological features			
Histologic grade			<0.001
III	6.128	2.322–16.172	
I or II	Reference		
Lymphovascular invasion			0.029
Present	2.931	1.116–7.696	
Absent	Reference		
Textural features			
Mean using SSF5 ^a^			0.003
<62.59 HU	4.714	1.685–13.183	
≥62.59 HU	Reference		
Entropy using SSF2			0.036
>5.31	2.770	1.068–7.184	
≤5.31	Reference		

^a^ SSF, spatial scale filter.

**Table 5 diagnostics-13-03569-t005:** Textural features according to clinical-pathological prognostic factors.

Features	Mean Using SSF5 ^a^		Entropy Using SSF2	
	Mean Value	*p*	Mean Value	*p*
Histological type		0.684		0.266
Invasive ductal	80.00 ± 51.24		4.95 ± 0.43	
Others	76.83 ± 39.70		5.04 ± 0.44	
Histological grade		0.967		0.002
I or II	79.94 ± 51.41		4.91 ± 0.43	
III	79.66 ± 45.50		5.09 ± 0.41	
Tumor size		<0.001		<0.001
≤2 cm	87.70 ± 55.38		4.82 ± 0.42	
>2 cm	67.43 ± 35.72		5.21 ± 0.32	
LN ^a^ status		0.347		0.040
Negative	81.85 ± 52.22		4.93 ± 0.44	
Positive	75.79 ± 44.22		5.04 ± 0.40	
Lymphovascular invasion		0.189		0.004
Absent	81.56 ± 50.19		4.93 ± 0.43	
Present	70.09 ± 45.80		5.15 ± 0.37	
ER ^b^ status		0.830		0.153
Negative	79.50 ± 50.28		4.95 ± 0.43	
Positive	81.10 ± 47.76		5.04 ± 0.41	
PR ^c^ status		0.497		0.166
Negative	78.58 ± 50.35		4.94 ± 0.42	
Positive	83.33 ± 47.88		5.03 ± 0.44	
HER2 ^d^ status		0.784		0.015
Negative	78.33 ± 41.70		5.10 ± 0.39	
Positive	80.23 ± 51.47		4.93 ± 0.43	
Molecular subtype		0.450		0.142
Luminal-like	79.81 ± 52.24		4.92 ± 0.43	
HER2-like	78.33 ± 41.70		5.10 ± 0.39	
Basal-like	82.04 ± 48.65		4.97 ± 0.42	
Ki-67 status		0.872		0.004
Low (<14%)	79.46 ± 48.07		4.91 ± 0.43	
High (≥14%)	80.51 ± 52.37		5.06 ± 0.40	

^a^ LN, lymph node; ^b^ ER, estrogen receptor; ^c^ PR, progesterone receptor; ^d^ HER2, human epidermal growth factor receptor type 2.

**Table 6 diagnostics-13-03569-t006:** Results of linear logistic regression analysis for entropy using SSF2.

Features	β (SE ^b^)	Odds Ratio	95% CI	*p*
Histologic type ^†^	−0.009 (0.070)	−0.135	−0.147–0.128	0.893
Histological grade ^‡^	0.039 (0.062)	0.627	−0.083–0.161	0.531
Tumor size ^††^	0.358 (0.054)	6.598	0.251–0.465	<0.001
LN status ^§^	0.025 (0.055)	0.451	−0.083–0.132	0.652
Lymphovascular invasion *	0.054 (0.074)	0.731	−0.092–0.200	0.466
HER2 ^a^ status ^#^	−0.075 (0.062)	−1.195	−0.197–0.048	0.233
Ki-67 status ^√^	0.111 (0.057)	1.949	−0.001–0.223	0.049

^†^ Independent variable was histological type with grouping invasive ductal vs. others. ^‡^ Independent variable was histological grade with grouping 1 and 2 vs. 3. ^††^ Independent variable was tumor size with grouping <2 cm vs. ≥2 cm. ^§^ Independent variable was LN status grouping negative vs. positive. * Independent variable was lymphovascular invasion grouping absence vs. presence. ^#^ Independent variable was HER2 status grouping negative vs. positive. ^√^ Independent variable was Ki-67 with grouping <14% vs. ≥14%. ^a^ HER2, human epidermal growth factor receptor type 2. ^b^ Standard error of the estimate.

## Data Availability

The data that support the findings of this study are available on request from the corresponding author (
S.E.S.).

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
