# Peer review of "Tumor Heterogeneity of Breast Cancer Assessed with Computed Tomography Texture Analysis: Association with Disease-Free Survival and Clinicopathological Prognostic Factor"

_diagnostics, 2023, doi:10.3390/diagnostics13233569_

Round 1

Reviewer 1 Report

Comments and Suggestions for Authors

Authors investigated breast cancer CT characteristics including a texture analysis in relation to prediction of disease-free survival and prognostic factors. The aim, methods and results are clearly presented.

I have minor suggestions:

1. Abstract - abbreviation CTTA is written twice

2. Authors used postcontrast CT scans to measure mean attenuation of breast cancer. Contrast enhancement depends on many factors and it is very individual. Can you clarify the possible effect of different patient and other characteristics that influence the contrast enhancement on the results of present study?

3. Delete line 397-399 "6. Patents....."

4. References 46 i 47 (in Limitation sections) are not listed in the Reference list.

Author Response

Thank you very much for taking the time to review this manuscript. Please find the detailed responses below and the corresponding revisions in track changes in the re-submitted files.

1. Abstract - abbreviation CTTA is written twice.

--> We deleted CTTA written twice in the abstract. 

2. Authors used postcontrast CT scans to measure mean attenuation of breast cancer. Contrast enhancement depends on many factors and it is very individual. Can you clarify the possible effect of different patient and other characteristics that influence the contrast enhancement on the results of present study?

--> Thank you for your comment. As you know, contrast enhancement can depend on many individual factors such as age, sex, body weight and height, cardiac output, renal function and hydration status. So, we added the sentence in the section of limitation as follows; Fifth, contrast enhancement of breast cancer on post-contrast chest CT scan can depend on many individual factors such as age, sex, body weight and height, cardiac output, renal function and hydration status, resulting in variability of CTTA [47].

3. Delete line 397-399 "6. Patents....."

--> We deleted patents section.

4. References 46 i 47 (in Limitation sections) are not listed in the Reference list.

--> Thank you for your meticulous comment. When we checked, reference 47 was same to reference 45. So, we changed reference number.

Reviewer 2 Report

Comments and Suggestions for Authors

The authors described the use of texture analysis for clinicopathological and outcome analysis, some interesting associations were found with DFS, but the manuscript requires some improvement before recommendation for publication.

- Only case selection criteria should be mentioned in the methods. The actual case numbers should be described in the results. The rationale for certian inclusion or exclusion criteria should be explained in the discussion.

- No criteria was mentioned for "breast surgery", either all cases should have received comparable surgery (all mastectomies, and BCTs/wide local excisions excluded), or all had complete excisions (clear margins), or margin status was included in the cohort and included in multivariable analysis.

- Correlational analysis between texture analysis and histopathological parameters should be performed.

- Staging is a key parameter in outcome analysis and cannot be omitted. Please revise.

Comments on the Quality of English Language

Line 107 (XXX)

Author Response

Thank you very much for taking the time to review this manuscript. Please find the detailed responses below and the corresponding revisions in track changes in the re-submitted files. 

1. Only case selection criteria should be mentioned in the methods. The actual case numbers should be described in the results. The rationale for certian inclusion or exclusion criteria should be explained in the discussion.

--> According to your comment, we moved the actual case numbers from materials and methods section to result section. We also added the rationale for exclusion criteria.

2. No criteria was mentioned for "breast surgery", either all cases should have received comparable surgery (all mastectomies, and BCTs/wide local excisions excluded), or all had complete excisions (clear margins), or margin status was included in the cohort and included in multivariable analysis.

--> According to your comment, we added the surgery type as follow; Of the 256 patients (mean age, 54 ± 11 years; range, 30–87 years), 176 (68.8%) patients underwent breast-conserving surgery and 80 (31.2%) patients underwent mastectomy. In addition, we also added the content about clear margin in the manuscript as follows:  

One patient who underwent breast-conserving surgery had positive resection margin at pathologic report.

As the number of patient who had positive resection margin was just one, margin status cannot be included  included in multivariable analysis.

3. Correlational analysis between texture analysis and histopathological parameters should be performed.

-->We performed linear logistic regression analysis using texture feature of entropy using SSF2 and histopathologic parameter and summarized it in the table 6. As you can see, entropy using SSF2 showed correlations with tumor size and Ki-67 status.

4. Staging is a key parameter in outcome analysis and cannot be omitted. Please revise.

--> According to your comment, staging was added in the univariate cox proportional hazards analysis. However, it didn't show a statistical significance. You can check it in the Table 3.